# Delayed Speech Perception and Production after Cochlear Implantation in Bilingual Children from Non-Native Families

**Nader Nassif [1], Maria Grazia Barezzani [2] and Luca Oscar Redaelli de Zinis [1,3,*]**

[1] Pediatric Otolaryngology Head Neck Surgery Unit, Children Hospital—ASST Spedali Civili of Brescia,
25121 Brescia, Italy; nadernassif@alice.it

[2] Unit of Pediatric Audiology and Phoniatrics, Children Hospital—ASST Spedali Civili of Brescia,
25121 Brescia, Italy; mariagrazia.barezzani@asst-spedalicivili.it

[3] Section of Audiology, Department of Medical and Surgical Specialties, Radiological Sciences,
and Public Health, University of Brescia, 25121 Brescia, Italy

[*] Correspondence: luca.redaellidezinis@unibs.it

**Abstract:** The aim of the study was to evaluate the outcomes of cochlear implantation (CI) in a group of immigrant deaf children living in a foreign language family, following up to 3 years of a personalized habilitation program compared to age-matched Italian CI recipients. Tests of speech perception ability such as the IT-MAIS, the LiP, the CAP, and speech production such as the MUSS have been used before CI and then after 6 months, 1 year, 2 years, and 3 years. Nonparametrical tests were chosen for comparison. Eight bilingual CI recipients were included in the study and matched to 11 Italian CI recipients. The difference between chronological age at implantation, age at diagnosis, hearing age, and verbal age in the two groups of children was not significant. Comparison of the auditory perceptive and linguistic abilities between the two groups showed significant differences only in preoperative MAIS and postoperative CAP (1 to 3 years). In agreement with other studies, we achieved good performances from bilingual children with CI and our personal experience confirm the attitude of promoting bilingualism throughout the rehabilitation process.

**Keywords:** deafness; cochlear implants; bilingualism; rehabilitation

## 1. Introduction

The benefits of early second language acquisition have been often disputed or seen with skepticism by some educators; most believe it might impair the correct learning of the first language. If this assumption were true, it might negatively affect the outcomes of cochlear implantation (CI) in deaf children who grow up in a bilingual environment. Current migratory trends of the populations from underdeveloped countries have led to a significant influx of foreign workers toward European countries. A foreseeable consequence is that families with deaf or hard of hearing children would follow the migrant worker in order to seek help; their health needs must therefore be addressed by the receiving institutions.

Offering a CI to a foreign family can prove a challenging task; [1] raising a child with a CI who needs to acquire two languages at the same time increases the odds of unsatisfactory results in language proficiency [2].

In addition, the issue of the type of language strategy to be applied in the CI habilitation program and the communication, socio–cultural, religious, and logistic problems with the foreign family represent a formidable hindrance.

The present study was conducted in one of the most industrialized cities in Northern Italy. The last two decades have witnessed the highest Italian immigration rate from developing countries. Such phenomenon has led to an exponential growth of the pediatric population seeking help for hearing difficulties. The Italian legislation guarantees both native family's children (NFC) and non-native family's children (NNFC) with hearing difficulties the same rights to access the National Health System. This implies that they can obtain free hearing aids, CI, and adequate rehabilitation. In our department, over one third of pediatric patients who received a CI belong to NNFC and live in bilingual homes.

The aim of this study was to determine the outcomes of CI in a group of immigrant deaf children living in a foreign language family, following up to 3 years of a personalized habilitation program, and to compare them to those achieved by age-matched NFC CI recipients, in order to test the hypothesis that bilingual CI recipients would never reach the same speech perception skills of NFC.

## 2. Materials and Methods

A prospective cohort longitudinal study was conducted among a group of pediatric cochlear implants recipients at the Pediatric Otorhinolaryngology Department of Brescia University Hospital between 2003 and 2011. They had all received the same brand of implant (Cochlear Co.; Lane Cove, Australia) and were followed for rehabilitation by the same speech therapist. Two groups were identified: One composed of NFC and the other of NNFC who lived in a bilingual environment.

Preoperatively, all children had undergone a battery of audiological tests including EOAE, ABR, behavioral audiometry, and videorecording of communicative skills. All were profoundly deaf and showed minimal speech perceptive abilities with hearing aids. Receptive and expressive language competence measures were obtained at the time of admission and after 3 months of hearing aid use, in order to properly assess the indication to CI, with multiple validated tests and questionnaires. They included tests of speech perception ability such as the IT-MAIS (Infant–Toddler Meaningful Auditory Integration scale), [3] the LIP (Listening-in-Progress Profile), [4] the CAP (Categories of Auditory Performance), [5] and speech production such as the MUSS (Meaningful Use of Speech Scale) [6]. All tests were delivered in Italian with a trained interpreter for each family language. The translation service was provided by the hospital through hired professional interpreters who previously received specific training on the subject. The interpreters assisted the parents throughout the post-CI habilitation sessions; one or both parents were also involved in developing pictorial and text material in their mother tongue with Italian translation, and personally assisted in the training sessions.

The strategy adopted for language rehabilitation was the audio–verbal therapy (AVT) approach. It consisted of bi-weekly sessions of "learning to listen", in which one or both parents were asked to attend and join the habilitation program, driven by the speech therapist.

Within the CI candidacy workup, a neuropsychological evaluation of the child paralleled the imaging studies (CT scan of temporal bones + MR of brain and ear) and the genetic investigations. A cognitive evaluation was performed by means of the Leiter's and Griffith's scales by experienced Pediatric Neuropsychiatrists. The level of social integration, communication capability and family awareness were assessed by a Psychologist trained in dealing with hearing impaired individuals.

The communicative skills in the Italian language of both parents were also tested and recorded, together with their level of knowledge of the acquired language, their alphabetization degree, social integration, and degree of information about the CI and expectations. Every problem that the team encountered or that the family reported were recorded and analyzed.

The auditory perceptive and linguistic abilities were measured at pre-determined intervals during the follow-up period: At the start of treatment, after 6, 12, 24, and 36 months.

The two groups of NFC and NNFC were not randomly selected but they represented natural cohorts.

The aim of the study was to compare the auditory perceptive and linguistic abilities of the two samples of children and to verify if there was an improvement along the 3 years follow-up interval, and if the NNFC showed a greater degree of difficulty.

First, we observed the time needed to acquire a statistically significant improvement of baseline scores and compared it between the 2 cohorts; the hypothesis that the NNFC children experience a delay was checked thereafter.

The second hypothesis being verified was the existence of "temporal windows" during which the children are more sensitive and responsive to the treatment; this objective was tested separately for each cohort, by comparing the scores obtained at contiguous couples of follow–up measures.

Data were analyzed by means of the statistical package SPSS. Non-parametrical tests were chosen for the verification of the hypotheses due to the low number of subjects analyzed.

The Mann–Whitney U test was used to compare the two groups for chronological age at the time of CI, age at the time of diagnosis, relative hearing age and relative verbal production age; it was also used to compare the auditory perceptive and linguistic abilities.

The F test of Friedman was selected in order to compare scores between baseline and follow-ups; the T test of Wilcoxon was chosen to compare scores at specific points along the follow-up timeline. The significance level was set at $p = 0.05$ for the F test of Friedman. Multiple comparison by means of the T test of Wilcoxon were evaluated at 1/4 of the nominal significance level, i.e., at 0.0125, in order to avoid a random significant result, being 4 the multiple comparison.

According to the designed non-parametrical tests, descriptive statistics include median (50th percentile) and interquartile range (IQR) (25th–75th percentile), as indicators of dispersion.

## 3. Results

Eighty-two children had undergone CI at the Pediatric Otorhinolaryngology Department of Brescia University Hospital between 2003 and 2011. Twenty-eight (34%) belonged to NNFC Italian families. Their age ranged from 8 months to 13 years. Their families had arrived from 12 different countries (8 Romania, 5 Albania, 2 Pakistan, 2 India, 2 Bangladesh, 2 Ukraine, 1 Egypt, 1 Argentina, 1 Brazil, 1 Morocco, 1 Senegal, 1 Tunisia,) and belonged to 14 different language roots. In most cases the diagnosis had been already established or suspected in their homeland; in less than half of the cases the family migrated to Italy in search of appropriate management of their child's deafness. There were 16 males and 12 females.

Among the 28 bilingual CI recipients, 8 were included in the study to match 11 NFC CI children. According to the audiological indications, 5 received a unilateral CI and maintained a contralateral hearing aid (bimodal stimulation). Three others received a bilateral implant in a sequential mode; the interaural interval time between the two implants is reported in Table 1. All NNFC children except one underwent a bilateral CI, either simultaneous ($n = 5$) or sequential ($n = 2$). The model of the implanted device and of the external speech processors are detailed in Table 1. The etiology of deafness was unknown in 5 NFC and 4 NNFC patients. In 2 NFCs and 2 NNFCs a mutation of the Connexin 26 genes were detected. One foreign child was deafened by meningitis at the age of 11 months. Two NFC patients were syndromic (1 Waanderburg and 1 CHARGE) and 1 NFC was considered syndromic based on his phenotypic appearance, although a known syndrome could not be resolved (Table 1).

The language spoken at home was the original mother tongue in all families (Table 2). The parental competence for Italian language was none in 3 cases, poor in 1, very limited in 2, sufficient in 8, and good in 2; none of the adult members of the immigrant families were fluent in Italian at the time of the CI. The two mothers from Pakistan and Morocco did not attend any school and showed absent or very limited understanding of the Italian language. Their husbands were only slightly better in speech understanding. Nine of the 10 parents of 5 Romanian children showed a sufficient level of understanding and communication ability in Italian; 2 of the parents, with a higher degree of education (secondary school) had achieved a good comprehension and fluency in Italian before their child's CI.

**Table 1.** Etiology of deafness and type of cochlear implantation (CI) in the two groups.

| | NFC | | | | | | NNFC | | | | |
|---|---|---|---|---|---|---|---|---|---|---|---|
| Patient | Unilateral vs. Bilateral CI | Delay (mo) | Speech Processor | Implant | Etiology | Patient | Unilateral vs. Bilateral CI | Delay (mo) | Speech Processor | Implant | Etiology |
| 1 | B sim | | CP 810 | N5 | Syndr | 12 | B seq | 31 | BTE | CI24RE | I |
| 2 | U | | BW | CI24RE | I | 13 | B seq | 28 | BW | CI24RE | CX26 |
| 3 | B seq | 19 | BW | CI24RE | I | 14 | B sim | - | Freedom BTE | CI24RE | CX26 |
| 4 | U | | BW | CI24RE | I | 15 | B sim | - | CP 810 | N5 | I |
| 5 | U | | BW | CI24RE | CHARGE | 16 | B sim | - | CP 810 | N5 | I |
| 6 | B seq | 9 | BW | CI24RE | I | 17 | U | - | Freedom BW | CI24RE | Consanguineity |
| 7 | U | | BTE | CI24RE | CX26 | 18 | B sim | - | Freedom BTE | CI24RE | I |
| 8 | B seq | 38 | BW | CI24RE | I | 19 | B sim | - | Freedom BTE | CI24RE | Meningitis |
| 9 | U | | BTE | CI24RE | Waanderburg | - | - | - | - | - | - |
| 10 | B sim | | CP 810 | N5 | I | - | - | - | - | - | - |
| 11 | U | | BW | CI24RE | CX26 | - | - | - | - | - | - |

NFC: native family's children; NNFC: non-native family's children; U: unilateral; B: bilateral; Seq: sequential; Sim: simultaneous; BTE: behind-the-ear; BW: body-worn; I: idiopathic; Syndr: unknown syndrome.

**Table 2.** Demographic features of the selected subgroup of non-native family's children (NNFC) CI.

| Patient | Country of Origin | Country of Birth | Home Language | Parental Competence for Italian Language | Parental Level of Education |
|---|---|---|---|---|---|
| 12 | Albania | Albania | Albanian | F: Sufficient<br>M: None | F: Primary<br>M: Primary |
| 13 | Romania | Italy | Romanian | F: good<br>M: Sufficient | F: Secondary<br>M: Primary |
| 14 | Romania | Italy | Romanian | F: Sufficient<br>M: Very limited | F: Primary<br>M: Primary |
| 15 | Romania | Italy | Romanian | F: Sufficient<br>M: Sufficient | F: Primary<br>M: Primary |
| 16 | Romania | Italy | Romanian | F: good<br>M: Sufficient | F: Secondary<br>M: Primary |
| 17 | Pakistan | Italy | Urdu | F: Very limited<br>M: None | F: None<br>M: None |
| 18 | Romania | Italy | Romanian | F: Sufficient<br>M: Sufficient | F: Primary<br>M: Secondary |
| 19 | Morocco | Morocco | Arabic | F: Poor<br>M: None | F: Primary<br>M: None |

F = Father; M = Mother.

The level of education was low (primary school) in 10 of 16 parents and intermediate in 3; both parents of the Pakistani family had received no education at all in their homeland. Mothers had a lower level of education compared to their husbands the majority of cases; 2 of 8 did not attend any school (Table 2).

In almost all cases the fathers had arrived in Italy alone in search of a job; once stabilized, they were joined by their wives. All children except two (1 Albanian and 1 Moroccan) where born in Italy, and the diagnosis of deafness was assessed at our Department. In the other two cases the diagnosis had been already established in their country of origin and hearing aids had been adopted; the CI had been suggested but was not affordable for the family, who decided to move to Italy where the health system is free and the CI is provided through hospitals at no cost for patients, when regularly registered.

The difference between chronological age at implantation, age at diagnosis, hearing age (speech perception abilities observed at the time of the CI), verbal age (speech production abilities observed at the time of the CI) in the two groups of children was not significant (Table 3).

**Table 3.** Median and interquartile range (IQR, 25th–75th percentile) of chronological age at implantation, age at diagnosis, age-related speech perception (hearing age), age-related speech production (verbal age) relative to normative standards for the Italian language.

| Analyzed Ages | NFC | NNFC | P |
|---|---|---|---|
| Age at CI (mo) | 48 (38–59) | 57 (38–80.5) | 0.5 |
| Age at diagnosis (mo) | 6 (3.5–17.5) | 18.5 (8–34.5) | 0.1 |
| Hearing age (mo) | 28 (22–35) | 26 (22–43) | 0.9 |
| Verbal age (mo) | 35 (30.5–46.5) | 36 (26.5–52) | 0.9 |

NFC: native family's children; NNFC: non-native family's children.

Comparison of the auditory perceptive and linguistic abilities between the two groups are reported in Table 4. Significant differences in preoperative MAIS and postoperative CAP (1 to 3 years) are observed.

**Table 4.** Median and interquartile range (IQR, 25th–75th percentile) of the auditory perceptive and linguistic abilities between the two groups.

| Questionnaire | NFC | NNFC | P |
|---|---|---|---|
| LIP (preoperative) | 5 (2–8) | 1.5 (0–2.5) | 0.051 |
| LIP (6 mo) | 30 (27.5–33.5) | 22 (20.5–34.5) | 0.4 |
| LIP (12 mo) | 38 (36–42) | 32 (29.5–41) | 0.1 |
| LIP (24 mo) | 42 (42–42) | 42 (38.5–42) | 0.4 |
| LIP (36 mo) | 42 (42–42) | 42 (41–42) | 0.4 |
| MAIS (preoperative) | 10 (7.5–14.5) | 1 (0–1.5) | 0.001 |
| MAIS (6 mo) | 29 (24–32) | 25 (16.5–33.5) | 0.6 |
| MAIS (12 mo) | 36 (35.5–38) | 35.5 (26.5–38) | 0.4 |
| MAIS (24 mo) | 40 (39.5–40) | 38.5 (35–40) | 0.1 |
| MAIS (36 mo) | 40 (40–40) | 40 (39.5–40) | 0.4 |
| CAP (preoperative) | 0 | 0 | 1 |
| CAP (6 mo) | 2 (2–3) | 2 (1.5–2) | 0.09 |
| CAP (12 mo) | 4 (3–5) | 3 (2.5–3) | 0.02 |
| CAP (24 mo) | 6 (5–6) | 4.50 (4–5) | 0.009 |
| CAP (36 mo) | 7 (7–7) | 5 (4.5–6.5) | 0.007 |
| MUSS (preoperative) | 3 (0.5–8.5) | 5.5 (1.5–6.5) | 0.6 |
| MUSS (6 mo) | 14 (10.5–20) | 13.5 (10–16.5) | 0.7 |
| MUSS (12 mo) | 20 (14–25) | 18.5 (15.5–21) | 0.8 |
| MUSS (24 mo) | 35 (29–39) | 27.5 (24–33) | 0.1 |
| MUSS (36 mo) | 40 (40–40) | 38 (37–40) | 0.08 |

NFC: native family's children; NNFC: non-native family's children; LIP: Listening-in-Progress Profile; MAIS: Meaningful Auditory Integration Scale; CAP: Categories of Auditory Performance; MUSS: Meaningful Use of Speech Scale.

The progression the auditory perceptive and linguistic abilities are displayed in Figure 1.

The F test of Friedman was always significant at a level of 0.0001 both for NFC and NNFC. Multiple comparison by means of the T test of Wilcoxon showed that the significant progression was mainly within the first two years in NNFC (Table 5).

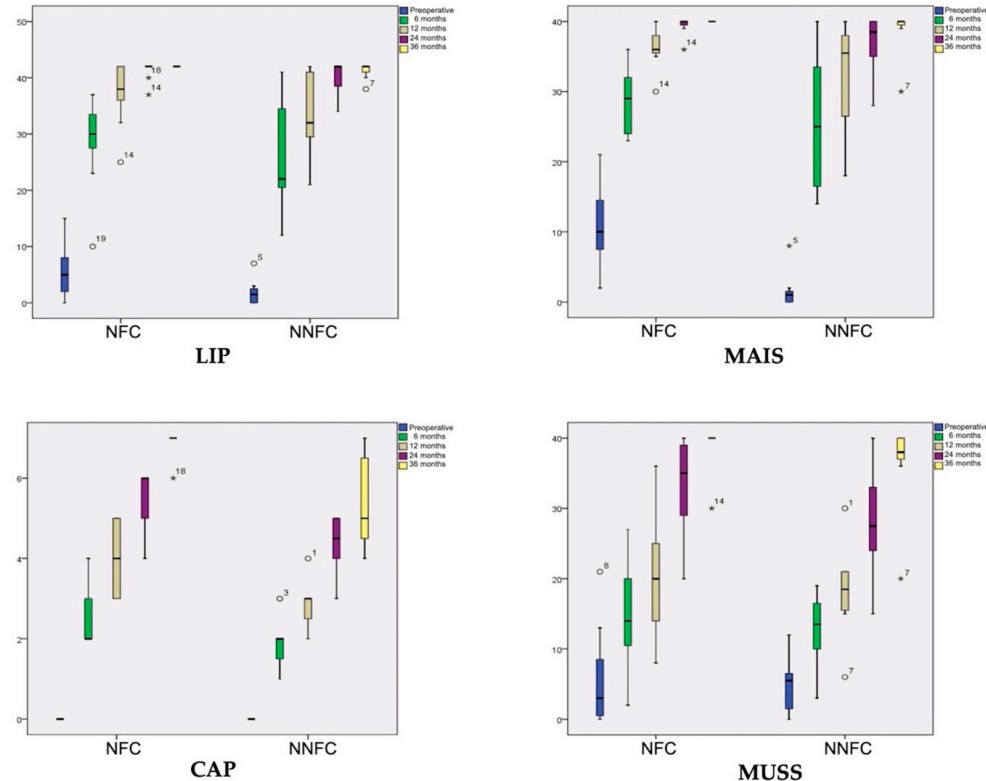

**Figure 1.** Scores of the auditory perceptive and linguistic abilities for the two populations at the different moments (the lower and upper dashes of the boxplot indicate the minimum and maximum observed values except for outliers, the box represents the interquartile range 25–75% and the bold dash inside the box is the median value, the circle represents the outliers and the asterisk represents the extreme values). LIP (Listening-in-Progress Profile), MAIS (Meaningful Auditory Integration Scale), CAP (Categories of Auditory Performance), MUSS (Meaningful Use of Speech Scale), NFC (native family's children), and NNFC (non-native family's children).

**Table 5.** P values for the multiple comparisons of the auditory perceptive and linguistic abilities between the two groups (significance level at 0.0125).

| Questionnaire | | NFC | NNFC |
|---|---|---|---|
| | | **P** | **P** |
| LIP (6 mo) | LIP (preoperative) | 0.005 | 0.012 |
| LIP (12 mo) | LIP (6 mo) | 0.003 | 0.012 |
| LIP (24 mo) | LIP (12 mo) | 0.02 | 0.02 |
| LIP (36 mo) | LIP (24 mo) | 0.2 | 0.1 |
| MAIS (6 mo) | MAIS (preoperative) | 0.003 | 0.012 |
| MAIS (12 mo) | MAIS (6 mo) | 0.003 | 0.02 |
| MAIS (24 mo) | MAIS (12 mo) | 0.005 | 0.02 |
| MAIS (36 mo) | MAIS (24 mo) | 0.1 | 0.04 |
| CAP (6 mo) | CAP (preoperative) | 0.003 | 0.01 |
| CAP (12 mo) | CAP (6 mo) | 0.002 | 0.02 |
| CAP (24 mo) | CAP (12 mo) | 0.003 | 0.01 |
| CAP (36 mo) | CAP (24 mo) | 0.002 | 0.04 |
| MUSS (6 mo) | MUSS (preoperative) | 0.003 | 0.012 |
| MUSS (12 mo) | MUSS (6 mo) | 0.003 | 0.011 |
| MUSS (24 mo) | MUSS (12 mo) | 0.003 | 0.011 |
| MUSS (36 mo) | MUSS (24 mo) | 0.012 | 0.02 |

NFC: native family's children; NNFC: non-native family's children; LIP: Listening-in-Progress Profile; MAIS: Meaningful Auditory Integration Scale; CAP: Categories of Auditory Performance; MUSS: Meaningful Use of Speech Scale.

## 4. Discussion

During the last decade the number of immigrants from underdeveloped countries toward Italy has increased dramatically. According to Italian Central Institute of Statistics the number of regular immigrants in January 2004 was 1,990,159, while in January 2011 it had more than doubled, reaching 4,570,317 units, on a total of 59,433,744 residents (7.5% of total residents) [7]. Brescia is one of the most industrialized cities in Italy; it has the highest mean concentration of immigrants (12.8% of the total population, and up to 26% in certain suburban areas) [7].

This high increase has led to an important demand on social and health care systems, with particular emphasis on language issues. In Italy, in order to benefit from the free health care support, an immigrant must be regularly registered to obtain a social security card. It allows him and his family the right and privilege to access the free National Health Service; as for deaf children, they can access all provisions available to the hearing disabled, such as hearing aids, hearing and speech therapy, and even a cochlear implant program and rehabilitation. Owing to the rapid demographic regional change, we had to update the health system at the Brescia University Hospital in order to respond to the new and challenging reality.

The incidence of deafness appears to be higher in immigrants compared to the NFC population. Based on the family's histories, we speculated that it could be, on one hand, the result of frequent number of marriages between relatives, and, on the other, to the poor hygienic, social, and health conditions during pregnancy, leading to frequent pre- and perinatal infections. Furthermore, we have observed that many immigrants, after stabilizing their residency in Italy, brought their families, including the hearing-impaired children, in order to seek help from a valid and free health system. In fact, in most homelands of the children's families, a valuable resource such as a cochlear implant is not readily available or its costs are not even minimally covered by the government institutions, constituting an unbearable economic burden for the family.

In our CI program for NNFC, we have faced and dealt with troublesome issues, such as communication with parents (often mediated by an interpreter), needing repeated interviews and counseling sessions with explanations given in a simplified fashion, gaining trust in the work of the team, and economical, logistic, cultural, religious, and job-related family life issues that must be addressed: our standardized treatment protocol had to be modified accordingly.

The most suitable and effective method of assessment of communicative performances and speech perception outcomes of CI in younger children is still a matter of debate. When bilingual children are the objects of analysis, the issue is even more relevant.

In the population studied, the development of auditory perception skills after CI was assessed by a battery of multiple tests including the MAIS, LIP, CAP, and MUSS, with the aim of comparing different approaches and try to eliminate biases.

The LIP evaluates the hearing ability in deaf children by assessing the listening level and sound identification. Our results with the LIP at 6- and 12-months follow-up showed a better but nonsignificant competence of discrimination and identification for NFC with respect to NNFC (Table 4). This performance gap was gradually bridged at 24 and 36 months (Table 4). We can affirm that the NFC develop the ability of discrimination and recognition about 6 months earlier the NNFC, who, nevertheless, catch up later on.

The MAIS questionnaire focuses on the development of the auditory milestones; it monitors the use of sensory auxiliary inputs and the reactions of the child to sound stimuli. In our study, the trend of differences between the 2 groups at the MAIS are similar to those observed at the LIP, owing to the fact that MAIS investigates exclusively the perceptual aspects, but ignores both language and contextual aspects, causing less disadvantage for the NNFC (Table 4).

The CAP questionnaire evaluates auditory performances in a way adopted for small children and is based on observation of the children and parental reports. We observed similar results up to 6 months, whereas significant differences were observed from 1 to 3 years in favor of NFC (Table 4).

The MUSS questionnaire evaluates oral language development represented by communication capability and the use of language in daily situations of any child wearing a hearing aid or a CI. In our study, the performance of the MUSS questionnaire denotes a progressive improvement of the expressive abilities of NNFC, despite language difficulties, with a nonsignificant lag from the control group until 36 months (Table 4).

Overall, analyzing the long-term results it can be observed that NNFC tend to achieve the same perceptive abilities of Italian children after 24 months of CI use; the trend is more evident at 36 months (Table 4). The initial gap, which is in the order of 6-12 months, was overcome after an adequate "immersion" in the acquired language outside the boundaries of the family, i.e., by socializing with Italian peers and educators at school, sports, and leisure activities.

We might argue that the scarce proficiency of immigrants' families in the use of the Italian language is negatively conditioning the expressive language acquisition by their children, leading to a restricted vocabulary use as well as lexical and syntactical shortcomings. This contributes to explain the persistent and significantly lower CAP scores and the initial lower scores at the MAIS and LIP compared to NFC CI users. MUSS scores are only slightly worst in NNFC even though almost all of them live mainly in a nonbilingual environment with a lower socio–cultural level. Often the NNFC produce an Italian language model similar to that of the parents.

Second language learning during childhood has been frequently viewed with skepticism, fearing that it may interfere with the first language acquisition and cause language impairment. To a certain extent, this is true also for normal hearing children, and thus educators and speech therapists are reluctant to accept a bilingual environment for children with a CI. Language learning capacity in normal hearing children is highly efficient in the first two years of life, which is considered the most important phase for linguistic development. Many studies have documented that in bilingual homes no special support is needed to master the two languages [8–11]. Early dual-language immersion in normal hearing children does not appear to impair language acquisition [12]. Frequent use of both languages might improve certain phonological skills, indicating that bilingualism can be beneficial for speech and language development [13]. In case of language difficulties in normal hearing bilingual children, they should be attributed to the child's innate capacity to acquire language, i.e., they could suffer a specific language impairment similarly to monolingual normal hearing children, and not to the consequence of the simultaneous acquisition of two languages [9,14–16].

Furthermore, individual differences in progress of language proficiency in hearing impaired bilinguals is conditioned both by the child's own abilities and by the IQ level that helps interpret poorly perceived words and compensate for missing acoustic information [17].

Last generation multichannel cochlear implants offer a broader access to the auditory signal with respect to hearing aids, due to higher degree of resolution of the phonetic features of spoken language. CI not only offers a better reception of sounds, but it also makes possible the incidental and natural language learning.

Age at implantation is a crucial factor for functional success of the implant. It is widely agreed that CI during the sensitive period for language learning permits the development of verbal skills similar to normally hearing children [15,18,19]. The same rule is valid for acquiring more than one language at the same time [18].

The first studies to examine the language abilities of bilingual children with CI come from the United States [8,15,19–24]. They showed that children developed competency in both languages, with the majority showing age-appropriate receptive and expressive language abilities in English, coming to the conclusion that learning another language could improve proficiency and vocabulary development in the second language [8,15,19–24]. Different results came from European institutions: In Germany, children growing up in bilingual homes performed significantly worse from monolingual homes and spoken German skills of the parents, integration of the family, and compliance with the habilitation program were critical in achieving sufficient language skills [25,26]. Similar observations come from France, where it was emphasized that bilingual children need intensive and good quality

input in both languages and parents should be encouraged to be involved in rehabilitation efforts [27]. A study from Spain showed considerable variability in the second language skills of bilingual participants, even though the parents of the bilingual participants reported that their children had second language skills that were either better than or at the level they had expected [28]. An Italian study showed that monolingual children obtained better results related to language skills, and they supposed that it is due to the status of bilingual families [29].

Our experience was similar to another Italian study [29]: The results obtained with implanted bilingual children initially showed slower progresses in auditory perceptive ability compared to their Italian peers. In the long term, the gap was progressively reduced but not completely eliminated. The rehabilitation program demanded more work and coordination within the staff and the territorial social agencies.

We had to insist with the foreign families for strict adherence to the scheduled program with the speech therapist as well as to homework. In some instances, we had to support the family with the help of institutional social assistance at home and transportation.

Despite our efforts, the results were inferior to matched monolingual deaf children with CI and possible explanations are:

- longer acquisition time for language comprehension and correct sentence construction;
- fluctuating performance due to absence from speech therapy sessions (e.g., during periods spent back in their homelands); and
- weak verbal memory and persistence of family linguistic model for a long time.

Other causative factors on outcomes of CI such as parental education and support, use compliance, mode of communication, and type of school and rehabilitation program attended, have been reported and ascribed in part to socioeconomically disadvantaged background [29,30], which is typical in immigrants from underdeveloped countries, as it is occurring in European countries.

Based on our current results and on the analysis of the literature [31], the optimal strategy to achieve satisfactory results with CIs in bilingual children should include:

- early diagnosis and application of hearing aid;
- optimal mapping of the CI (or CIs if binaural) and tailored fitting of the contralateral hearing aid (if present);
- good cooperation and regular involvement of the family;
- intensive and constant long-term speech therapy;
- regular exercise at home in own primary language; and
- stimulating linguistic environment at home, in social activities and educational setting.

## 5. Conclusions

Bilingualism in deaf children is an important recent issue with controversial aspects if a CI has to be considered. In agreement with other studies, we achieved good performances from bilingual children with CI and our personal experience confirms the attitude of promoting bilingualism throughout the rehabilitation process.

Early timing for implantation is essential for a good result, combined with intensive and sustained speech therapy. Special attention should be paid to social integration of the family, communicative improvement (i.e., parental learning of 2nd language), and strict adherence to follow-up in the postoperative period.

The use of the mother language should be encouraged in order to create a normal social and cultural familial background.

If feasible, in every CI center speech therapists and educators should be advised to adopt the bilingual code in their rehabilitation program, possibly through the mediation of an interpreter.

Long-term follow-up is necessary in order to achieve good results, which start to appear after 18 months post CI activation.

**Author Contributions:** M.G.B. was involved in selection and in rehabilitation of patients to be implanted, and L.O.R.d.Z. and N.N. carried out surgical interventions. All authors have made substantial contributions to the conception and design of the work, the acquisition, analysis, and interpretation of data, and in drafting the work and revising it. All authors have approved the submitted version and agree to be personally accountable for the author's own contributions and for ensuring that questions related to the accuracy or integrity of any part of the work are appropriately investigated, resolved, and documented in the literature. All authors have read and agreed to the published version of the manuscript.

**Funding:** This research received no external funding.

**Institutional Review Board Statement:** The study was conducted according to the guidelines of the Declaration of Helsinki and approved by the Ethics Committee) of ASST Spedali Civili Brescia (protocol code 0083787, date of approval 20th November 2019).

**Informed Consent Statement:** Informed consent was obtained from all subjects and parents involved in the study.

**Data Availability Statement:** The data presented in this study are available on request from the corresponding author. The data are not publicly available due to underage subjects.

**Conflicts of Interest:** The authors declare no conflict of interest.

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
