# Peer review of "Delayed Speech Perception and Production after Cochlear Implantation in Bilingual Children from Non-Native Families"

_2504-463X, doi:10.3390/ohbm2010004_

Round 1

Reviewer 1 Report

  1. General remark: please, pay attention to the possible role of sign language, which could be used (partially).
  2. Introduction: please, pay attention to the differences that could be found in the home situation of the children: immigrant parents and other family members could only speak their native language or not?!
  3. Method: The two groups of children consist of different numbers of children and because of the size of the group we should expect that the numbers are equal, but this is not the case (8 NNFCS vs 11 NFCS), why, please explain!
  4. Method: it is not clear in what respect deafness of parents could play a role, please pay attention to it, because our experience is that it happens sometimes that parents are deaf and those parents can play a different role with respect to the communication with their deaf children with or without CI.
  5. Results: please, could you explain a little bit more about the differences monaural CI vs bilateral CI.
  6. Results and discussion: in your study the bilingual children have 14 different language roots, but most of them originate from developed countries. In our country and we expect that in most western countries the majority of bilingual children originate from developing countries with difficult home country and family situations. Question: please, pay attention to this phenomenon because we expect that it could play a role, e.g. level of general physical and mental development, etc.

Author Response

The revised manuscript was checked by a native English speaker

General remark: please, pay attention to the possible role of sign language, which could be used (partially).

Sign language was never used

Introduction: please, pay attention to the differences that could be found in the home situation of the children: immigrant parents and other family members could only speak their native language or not?!

Italian language competences are reported in lines 149-158 and in Table 2. There are too many differences between the 8 families to compare data

Method: The two groups of children consist of different numbers of children and because of the size of the group we should expect that the numbers are equal, but this is not the case (8 NNFCS vs 11 NFCS), why, please explain!

This depends on selection criteria specified in lines 59-61: single brand cochlear implant and single speech therapist allowed to include 8 NNFCS and 11 NFCS

Method: it is not clear in what respect deafness of parents could play a role, please pay attention to it, because our experience is that it happens sometimes that parents are deaf and those parents can play a different role with respect to the communication with their deaf children with or without CI.

No parents were deaf

Results: please, could you explain a little bit more about the differences monaural CI vs bilateral CI.

We tried to analyze differences between monaural CI vs monoaural CI with contralateral hearing aid vs bilateral CI, but the numbers are too small to draw any meaningful conclusions.

Results and discussion: in your study the bilingual children have 14 different language roots, but most of them originate from developed countries. In our country and we expect that in most western countries the majority of bilingual children originate from developing countries with difficult home country and family situations. Question: please, pay attention to this phenomenon because we expect that it could play a role, e.g. level of general physical and mental development, etc.

We reported the level of education of the parents in lines 149-158 and in Table 2. They cannot be compared with those of Italian families.

Reviewer 2 Report

I don't think the patients' initials are appropriate ethically, because they can be identified.

No additional comments.

Author Response

The revised manuscript was checked by a native English speaker We removed the initials of patients as you suggested